# Facile Synthesis of 3D Interconnected Porous g-C$_3$N$_4$/rGO Composite for Hydrogen Production and Dye Elimination

Congyue Zhao [1], Hengchao Sun [2,*], Chunling Li [1], Manrong Wang [2], Jiahang Wu [2], Minghui Chen [1], Shuai Jiang [2], Tianqi Niu [1] and Dong Liu [1,*]

[1] School of Public Health, Xinxiang Medical University, Xinxiang 453003, China; zcy19990518@163.com (C.Z.); li_chunling163@163.com (C.L.); qq1942767431@163.com (M.C.); tianqi_niu1990@126.com (T.N.)

[2] Beijing Smart-Chip Microelectronics Technology Co., Ltd., Beijing 100192, China; wangmanrong@sgchip.sgcc.com.cn (M.W.); wujiahang@sgchip.sgcc.com.cn (J.W.); jiangshuai@sgchip.sgcc.com.cn (S.J.)

\* Correspondence: sunhengchao@sgchip.sgcc.com.cn (H.S.); liudong081@163.com (D.L.); Tel.: +86-373-3831063 (D.L.)

**Abstract:** Photocatalytic materials can effectively decompose water to produce hydrogen and degrade pollutants, ameliorating environmental issues. These materials are currently a popular research topic for addressing energy shortages and water pollution issues worldwide. Herein, we prepared composite catalysts with g-C$_3$N$_4$/rGO heterojunctions formed via the stacking of reduced graphene oxide (rGO) nanosheets and three-dimensional (3D) carbon nitride, and the catalysts displayed excellent photocatalytic activity in experiments for hydrogen production (4.37 mmol g$^{-1}$ h$^{-1}$) and rhodamine B elimination (96.2%). The results of structural characterization showed that the recombination of rGO has no effect on the morphology of g-C$_3$N$_4$, and the photochemical characterization results showed that the photogenerated electron migration of the prepared composite was accelerated. Additionally, a possible mechanism of enhancement involving synergy between the 3D structure of the catalyst and the g-C$_3$N$_4$/rGO heterojunctions was proposed on the basis of catalyst characterization and photocatalytic experiments. The prepared composite catalysts had large specific surface areas and abundant adsorption sites due to the 3D structure, and the g-C$_3$N$_4$/rGO heterojunction provided high electron mobility, resulting in low recombination of photoinduced electron and hole pairs and high conductivity. Moreover, free radical species that may play a substantial role in the photocatalytic process were analyzed via free radical quenching experiments, and possible catalytic mechanisms were presented in this study.

**Keywords:** environmental pollution; energy shortage; 3D structure; g-C$_3$N$_4$/rGO heterojunction; photocatalysis performance

## 1. Introduction

The rapid growth in the urban population and rapid economic development have caused a series of problems in recent decades, such as environmental pollution and energy shortages [1,2]. Due to the high-speed development of the social economy, the demand for traditional fossil fuels is increasing due to large-scale use in recent decades, which not only brings a variety of environmental problems, but also triggers energy crises on account of limited traditional fossil fuels [3]. Hence, photocatalytic decomposition of water to produce H$_2$ has become a promising method. Rhodamine B (RhB) is readily soluble in water and has good stability. As a result, it is widely used in the paper dyeing industry, including the dyeing of acrylic fabrics, the production of paints and as a fluorescent stain for cells [4,5]. However, RhB is also one of the components of industrial wastewater due to its numerous applications. In addition, discharges of industrial wastewater with RhB have had a strong impact on human health due to the carcinogenicity of RhB [6]. Therefore, it is necessary to address these toxic and harmful pollutants in water [7,8]. Water pollution treatment technologies are diverse, including the activated sludge process [9],

adsorption method [10] and oxidation technology [11]. Among them, photocatalysis has been widely studied by researchers due to its high efficiency of degradation, low energy consumption and non-secondary pollution generation [12,13]. At present, photocatalysis has shown many benefits for environmental remediation; for example, it can be used to degrade various contaminants [14] including broad-spectrum antibacterial agents in aqueous solutions [15] and $NO_X$, $SO_X$ and VOCs in the gas phase [16]. The preparation process for photocatalysts strongly affects photocatalytic activity, and good catalysts must exhibit a good light response. Thus, preparing and optimizing photocatalysts with high catalytic activity are important to address the above problems.

Nonmetallic catalysts have advantages such as low cost, environmental friendliness and ease of structural modification and have attracted widespread attention. Among these catalysts, graphitic carbon nitride ($g$-$C_3N_4$) is widely used by researchers due to its stable structure, simple preparation process and suitable band gap [17–19]. However, bulk $g$-$C_3N_4$ shows inadequate absorption of visible light, rapid recombination of photogenerated charge carriers and low quantum efficiency, which limits the photocatalytic activity of the catalysts [20]. To date, different tactics have been devised by researchers to increase photocatalytic efficiency of $g$-$C_3N_4$, including morphology modification [21], defect engineering [22] and heterojunction fabrication [23]. Morphology modification and heterojunction construction, achieved by combining two semiconductor materials, have been recognized as simple and effective approaches to improve the catalytic activity of $g$-$C_3N_4$ to achieve higher photoelectric conversion efficiency and lower energy loss [24–26].

Morphological modulation is an effective way to increase the specific surface area of catalysts [27]. Compared to nanotube structures and quantum dots, two-dimensional (2D) nanosheet structures have attracted widespread attention because they have large specific surface areas and abundant reactive sites [28]. Therefore, they are effective materials for prolonging the life of photogenerated charge carriers and enhancing the transfer charge carriers produced during photocatalysis [29–31]. However, the sheet structures of 2D $g$-$C_3N_4$ lead to stacking and adhesion, which reduces surface area and is unfavorable for photocatalytic reactions. In this study, a 3D structure assembled from 2D structures was used to solve the above problem perfectly. The prepared catalysts had larger surface areas and more active sites, which exhibited the excellent adsorption capacity of RhB, efficient utilization of incident illumination and rapid charge transfer due to the 3D interconnected structure [32]. Sheng et al. successfully prepared 3D $g$-$C_3N_4$ with cyanuric acid and melamine, and this catalyst showed enhanced adsorption and photocatalytic activity [33]. The comparison of photocatalytic performance with reported 2D $g$-$C_3N_4$-based materials and 3D rGO/$g$-$C_3N_4$ in this work were listed in Table S1. The $g$-$C_3N_4$ with a 3D porous structure exhibited better photocatalytic activity than a 2D structure. It follows that 3D structure are advantageous for the photocatalytic reaction to proceed.

Additionally, the heterojunction construction mentioned in this work is also an excellent way to optimize the structure of $g$-$C_3N_4$ [34,35]. The ultrathin structure of graphene leads to a large specific surface area, combined with its strong chemical stability and high electrical conductivity, making it an ideal material to compound with $g$-$C_3N_4$ [36]. The large specific surface area of $g$-$C_3N_4$/rGO synthetic catalysts can not only result in enhanced the adsorption of pollutants, but also increase the number of active sites for reaction [37]. In addition, there are a large number of functional groups on the surface of rGO that easily combine with organics, such as carboxyl groups and hydroxyl groups, which are also beneficial for the photocatalytic degradation of contaminants [38,39]. Moreover, rGO can promote photogenerated electron transfer at the interface between the neighboring semiconductors [40]. Hence, the stacking of rGO and 3D $g$-$C_3N_4$ to arrange $g$-$C_3N_4$/rGO heterojunctions is expected to further improve the photocatalytic activity of $g$-$C_3N_4$ due to its 3D porous structure.

In this work, we synthesized porous 3D $g$-$C_3N_4$ via a handy one-step dead burn method and synthesized $g$-$C_3N_4$/rGO composite catalysts via reducing graphene oxide (GO) under a nitrogen atmosphere and compounding it with $g$-$C_3N_4$. The microstructures

and structural characteristics were analyzed by structural and photoelectrochemical characterization tests. The photocatalytic production of hydrogen by decomposition of water and degradation of RhB was also carried out to evaluate the photocatalytic performance of the composite catalysts. Finally, we explored the possible mechanisms underlying the enhancement in photocatalytic efficiency on the basis of the above analysis and results.

## 2. Results

### 2.1. Microscopic Morphology and Structural Description

The copolymerization mechanism of 3D g-$C_3N_4$ and the constitution of the g-$C_3N_4$/rGO heterojunctions are elucidated in Figure S1. The self-assembly structures of melamine and cyanuric acid facilitated the formation of a 3D structure, which was due to the hydrogen bond interaction of MCS during stirring. On the other hand, the thermal instability of cyanuric acid led to the production of a large amount of gas during the calcination process, which is another significant condition for the 3D structure. Additionally, GO combined with g-$C_3N_4$ to form heterojunctions during the process of annealing and GO was reduced to rGO in the existence of nitrogen, resulting in g-$C_3N_4$/rGO heterojunctions. In the end, the 3D g-$C_3N_4$/rGO composite catalysts with hollow porous structures and g-$C_3N_4$/rGO heterojunctions were synthesized resoundingly.

Field emission scanning electron microscopy (FESEM, S-4800, Hitachi, Tokyo, Japan) and transmission electron microscopy (TEM, JEM-2010, JEOL Co., Ltd., Beijing, China) were conducted to analyze the microscopic morphology of the catalysts, and the results are shown in Figure 1. BCN presented a compact and bulky appearance, with anomalous aggregation, as shown in Figure 1a. As seen from Figure 1b, the shape of PCN appeared to consist of a 3D porous network structure, which differed from that of BCN. The 3D structure was formed by the crosslinking of 2D nanosheet, which occurred because supramolecular melamine–cyanuric acid was present during the calcination process. Similar to PCN, the GPCN catalysts emerged a hollow polyporous structure that tended to helicoid as depicted in Figure 1c–e, indicating that the loading of rGO flakes had no influence on the morphology of PCN, which was in favor of the absorption. The TEM image of PCN (Figure 1f) displayed stacked ultrathin g-$C_3N_4$ nanosheets, which primarily resulted in the generation of the 3D structure. Meanwhile, the ultrastructure of GPCN-2 showed the stacking of rGO nanosheets and g-$C_3N_4$, which enhanced the specific surface area of catalysts. However, the rGO nanosheets could not be clearly detected in the TEM diagram since the sample was sprayed with gold during the characterization. In addition, the distribution of the elements in the GPCN-2 catalyst was evaluated by energy dispersive spectroscopy (EDS, S-4800, Hitachi, Tokyo, Japan), as shown in Figure 1i, which corresponded to the analysis of elements mapping (Figure 1h), manifesting the uniform distribution of the C, N and O element. The consequence of EDS and elements mapping suggested the homogeneity in the compound catalysts.

The surface properties and pore diameter distribution of the catalysts were measured by $N_2$ adsorption–desorption isotherm from Figure 2a. Evidently, the synthesized samples exhibited isotherms of type III and a hysteresis loop corresponding to $H_3$ type, demonstrating that there were mesopores in the catalysts [41,42]. The special surface areas (SSA) and total pore volumes are listed in Table S2. PCN presented a much higher SSA of 80.604 $m^2 \, g^{-1}$ than BCN (19.738 $m^2 \, g^{-1}$), which resulted from the porous network structure, leading to outstanding adsorption capacity and catalytic activity [43]. What is interesting is that the SSA of GPCN catalysts was slightly higher than PCN, and that of GPCN-2 reached 90.909 $m^2 \, g^{-1}$, which indicated that the presence of rGO effectively increased the surface area of PCN and promoted the redox reaction [44]. Additionally, the total pore volumes of the samples showed a parallel trend, elucidating that the g-$C_3N_4$/rGO heterojunctions were beneficial to the adsorption of the catalysts. Moreover, the pore diameter distribution of catalysts was in agreement with SSA in Figure 2b. Hence, the integration of g-$C_3N_4$ and rGO was conducive to enhancing the capacity of adsorption and photocatalysis.

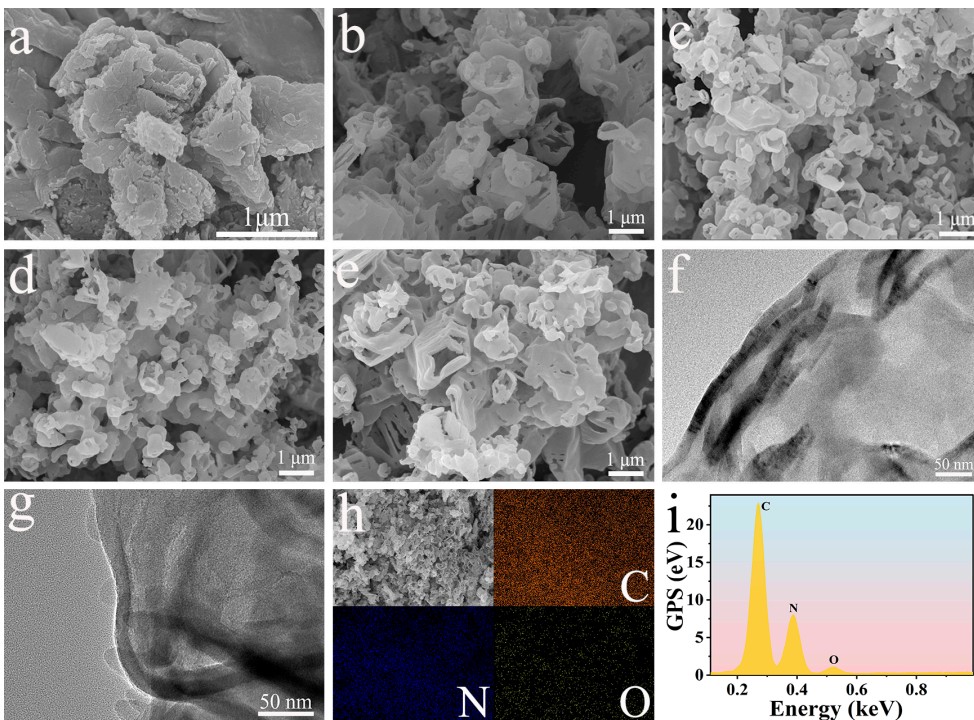

**Figure 1.** FESEM images of (**a**) BCN, (**b**) PCN, (**c**) GPCN-1, (**d**) GPCN-2, and (**e**) GPCN-3; TEM images of (**f**) PCN and (**g**) GPCN-2; (**h**) mapping and (**i**) EDS spectrum of GPCN-2.

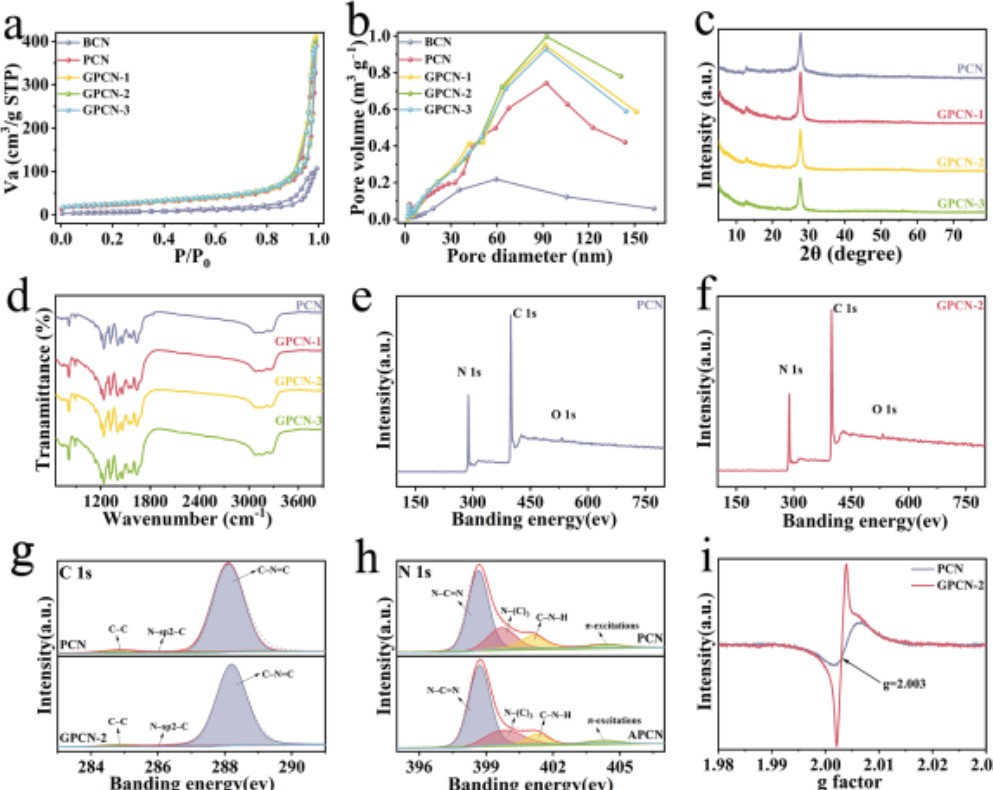

**Figure 2.** (**a**) N$_2$ absorption–desorption isotherms, (**b**) the pore size distribution, (**c**) FTIR spectra and (**d**) XRD patterns of the catalysts; (**e**) XPS survey spectra, high-resolution XPS spectra of (**f**) C 1s, (**g**) N 1s and (**h**) O 1s of PCN and GPCN-2; (**i**) EPR spectra of PCN and GPCN-2.

The structural features of the materials were measured by X-ray diffraction (XRD, PANalytical, EA Almelo, The Netherlands), as shown in Figure 2c. Two distinct diffraction peaks at 12.7° and 27.6° were found in the spectrum of PCN, which were attributed to the (100) and (002) crystal faces, caused by the accumulation of tri-s-triazine units and superposition of graphite layers, respectively [45]. Based on observation, the GPCN catalysts had XRD patterns similar to that of PCN, and showed no shift of peaks, confirming that rGO nanosheets caused no alteration in the crystalline structure of g-C$_3$N$_4$. The amount of graphene added was too small to allow for the observation of the characteristic diffraction peak of rGO [46]. For further exploration of the internal chemical bonds of the materials, Fourier transform infrared (FTIR, NEXUS 470, Nicolet, WI, USA) spectroscopy was detected just as Figure 2d illustrates. The spectra of the samples presented the same stretching vibrations, manifesting no remarkable variation in the structure of PCN, which mutually corroborates the results in XRD. Clearly, the peaks at 810 cm$^{-1}$ and 1200–1680 cm$^{-1}$ were related to the stretching vibration mode of triazine rings and C–N/C=N bonds [47,48]. Additionally, the existence of O-H bonds resulted in the apparent peak at 3000–3400 cm$^{-1}$ [47]. As expected, the establishment of g-C$_3$N$_4$/rGO heterojunctions did not change the original structure of the PCN, which is in good agreement with electron microscopic results.

In the meantime, the elements in the catalysts were qualitatively and quantitatively assessed via X-ray photoelectron spectroscopy (XPS, Axis Ultra, Kratos Analytical, Manchester, UK), the results of which are shown in Figure 2e–h. Figure 2e,f displays the survey XPS spectra of PCN and GPCN-2, correspondingly. The spectra manifested high intension of C and N elements, whereas the intensity of O element was low. This was attributed to the water adsorbed on the surface of the material and oxygen present in the environment [49]. As seen from the elemental composition in Table S3, the percentage of C in GPCN-2 increased to 41.98% compared to PCN (40.96%), and the C/N ratio aggrandized from 0.71 to 0.73, demonstrating the successful introduction of rGO nanosheets. The spectra of C 1s were divided into three typical peaks in PCN and GPCN-2 (Figure 2g). The carbon in the environment and C–C in rGO were assigned to the peak at 284.8 eV [50]. The peaks at 286.3 and 288.1 eV were assigned to sp2 hybrid carbon in the pyrimidine rings and C–N=C bonds [51]. A significant enhancement in the characteristic peak for rGO was not observed in GPCN-2 potentially because a very small amount of rGO was added. The N 1s spectra of PCN and GPCN-2 are displayed in Figure 2h, which were fitted with three characteristic peaks at 398–402 eV. The peak at 398.7 eV was ascribed to the sp2 of N hybridization in N–C=N bonds, the peak at 399.8 eV was due to N–(C)$_3$ and the peak at 401.3 eV was identified as sp hybridization of NH in the terminal amino groups. In addition, the π-bonds formed by the interaction of the π orbital in a nitrogen atom with the σ orbital of a neighboring atom led to a low-intensity peak at 404.3 eV [52]. The circumstance of unpaired electrons in the catalysts was evaluated by an electron paramagnetic resonance (EPR, MEX-nano, Bruker, Karlsruhe, Germany) test, and the outcomes are displayed in Figure 2i. PCN and GPCN-2 presented unambiguous Lorentzian lines with a g value of 2.003, which proved that the prepared catalysts contained single electrons. Noticeably, the EPR intensity of GPCN-2 was higher than that of PCN, which indicated more delocalized electrons and better photocatalytic activity [53].

## 2.2. Photology and Electrochemistry Peculiarity

The ability of the catalysts to absorb light at different wavelengths was estimated by UV-Vis diffuse reflection spectroscopy (DRS, U-3900, Hitachi, Tokyo, Japan). As depicted in Figure 3a, PCN and GPCN catalysts exhibited the same absorption spectrum with an absorption edge at approximately 450 nm. The similar absorption curves indicated that the addition of rGO had no effect on the photoabsorption capacity of the catalysts. The absorption capacity of rGO was the highest among all samples since graphene is black [54]. Furthermore, the band gap of the material was calculated based on Kubelka–Munk theory (Equation (1)), in which A, hν and α are a constant, photon energy and absorption coefficient, respectively [55]. The Eg values of PCN and rGO are displayed in

Figure 3b,c with the values of 2.57 eV and 1.44 eV, respectively. In addition, the Eg value of GPCN-2 is 2.53 eV, which was displayed in Figure S2. As is known to all, photocatalysis performance is closely related to the transplantation of photon-generated charges and the recombination of electron-hole pairs [56]. Electrochemical impedance spectroscopy (EIS) was measured to determine the charge separation efficiency of the samples, which is related to the performance of photocatalysis. As exhibited in Figure 3d, the EIS profiles of the catalysts are typical of the Nyquist plots, in which arcs form by the interaction of the constant phase element (CPE) and charge transfer resistance (Rct) [57]. The results show that the radius of GPCN-2 in the spectrogram is the smallest, which indicated that GPCN-2 showed the highest charge separation, resulting in superior photocatalytic capability [46]. Moreover, the photocurrent results of PCN and GPCN-2 are described in Figure S3, and are consistent with EIS results. The maximum intensity of GPCN-2 (16.8 μA) was 3.05-fold that of PCN (5.5 μA).

$$(\alpha h\nu)^2 = A(h\nu - Eg) \tag{1}$$

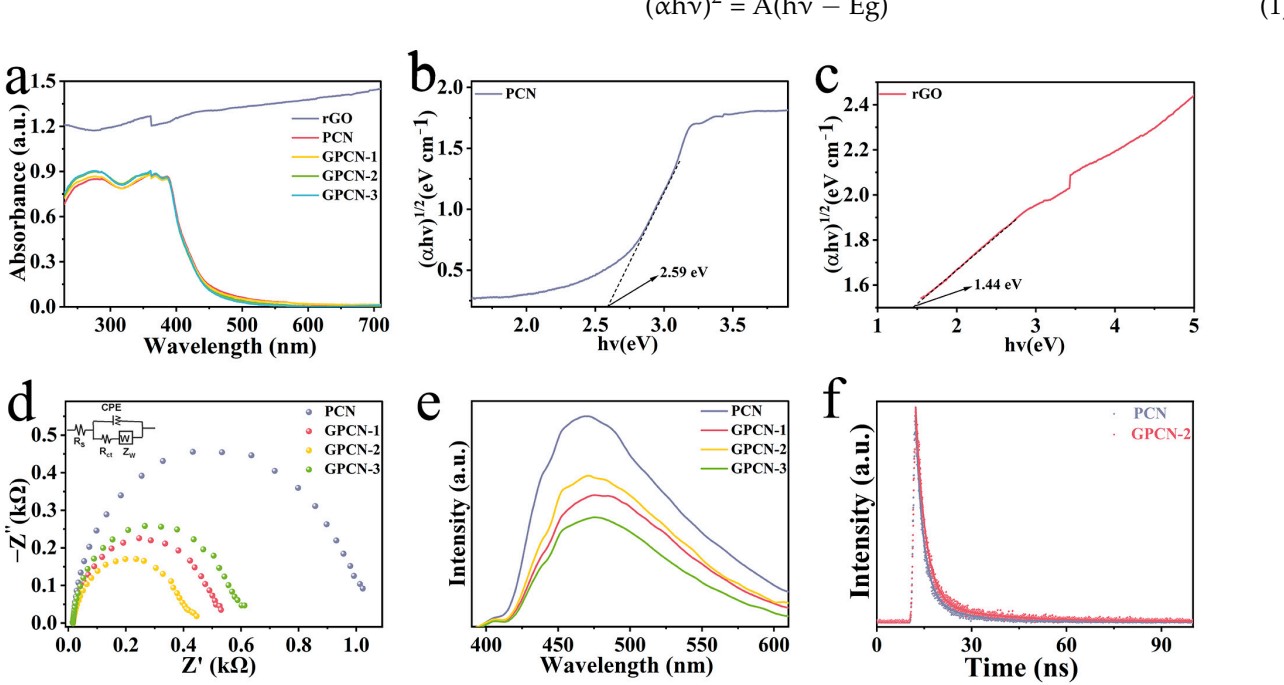

**Figure 3.** (**a**) UV-vis diffuse reflectance spectra of samples; the plots of transformed Kubelka–Munk function versus light energy of (**b**) PCN and (**c**) rGO; (**d**) electrochemical impedance spectroscopy and (**e**) steady state of as-synthesized samples; and (**f**) time-resolved PL spectra of PCN and GPCN-2.

Additionally, we conducted photoluminescence (PL, Fluoromax-4, HORIBA Jobin Yvon, Kyoto, Japan) and time-resolution photoluminescence (TRPL, FLSP920, EI, Edinburgh, UK) spectra to analyze the reorganization of photogenerated charge intermediaries. Under normal conditions, a low PL intensity represents low charge recombination [58]. It can be observed from Figure 3e that PCN appeared to have the highest intensity, whereas GPCN-2 presented the lowest intensity among all catalysts, which proved that the g-$C_3N_4$/rGO heterojunction contributed to a reduction in the recombination of electron and hole, improving photocatalytic activity [59]. The TRPL spectrogram of PCN and GPCN-2 were fitted in Figure 3f. The decay time ($\tau_x$) and relative amplitude ($A_x$) are listed in Table S4, and the mean fluorescence lifetime ($\tau$) was acquired through Equation (2) [60]. GPCN-2 had a mean fluorescence lifetime with a value of 3.45 ns, which is higher than that of PCN (2.56 ns). The higher mean lifetime elucidated that the excitation lifetime of photogenerated electric charge carriers increased, which was conducive to the combination of dissociative charges and active particles [61].

$$(\tau) = \frac{A_1\tau_1^2 + A_2\tau_2^2}{A_1\tau_1 + A_2\tau_2} \tag{2}$$

### 2.3. Photocatalytic Performance Evaluation

The hydrogen evolution experiments were conducted to evaluate the photocatalytic activity of the prepared samples, where triethanolamine acted as the sacrificial agent. Figure 4a exhibits the amount of hydrogen production under different illumination times with different samples. It can be observed that BCN presented the lowest hydrogen evolution of 0.35 mmol g$^{-1}$ due to its tight and massive structure. By comparison, the quantity of hydrogen production at 3 h of PCN was much higher, reaching 3.62 mmol g$^{-1}$ and was 10.34 times that of BCN. The results confirmed the consistent network interconnection structure, which facilitated the adsorption of water. More surprisingly, hydrogen production of GPCN catalysts all significantly improved compared with PCN, among which GPCN-2 achieved a hydrogen production amount of 13.11 mmol g$^{-1}$, 3.62 times that of PCN. As described, the affiliation of rGO nanosheets further increased the adsorption capacity of the catalysts and promoted the separation of photogenerated charges, which was to the benefit of improving the catalytic performance. The production rates of the different catalysts are displayed in Figure 4b. GPCN-2 (4.37 mmol g$^{-1}$ h$^{-1}$) exhibited the highest rate of H$_2$ release, which was 36.42 and 3.61 times that of BCN (0.12 mmol g$^{-1}$ h$^{-1}$) and PCN (1.21 mmol g$^{-1}$ h$^{-1}$). The consequence mutually corroborated the results in the amount of hydrogen production. However, the hydrogen production of GPCN-3 (10.87 mmol g$^{-1}$) decreased with the increase in rGO, which may be due to the reason that the excess rGO hindered the capture and absorption of light by g-C$_3$N$_4$ [3].

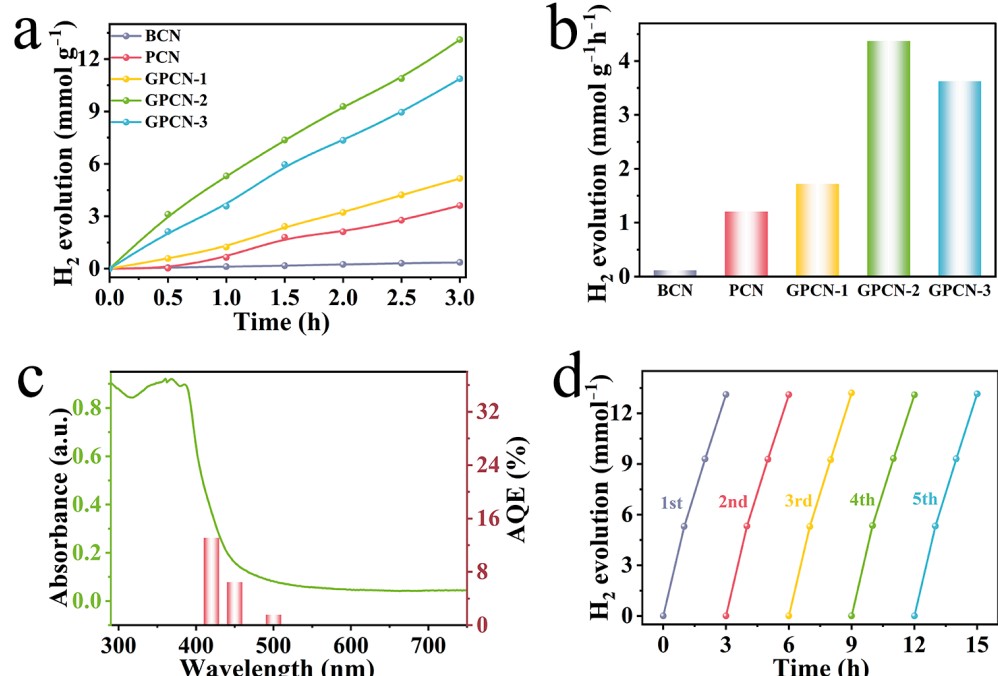

**Figure 4.** (**a**) H$_2$ evolution and (**b**) evolution rate of the catalysts; (**c**) wavelength dependent apparent quantum efficiency and (**d**) stability of H$_2$ evaluation by GPCN-2 catalysts.

Furthermore, the apparent quantum efficiency (AQE) of the catalysts was calculated via Equation (3), which is one of the important indexes to measure the activity of a photocatalytic reaction [62]. As depicted in Figure 4c, the AQE of GPCN-2 at 420 nm, 450 nm and 500 nm was 3.28%, 1.61% and 0.39%, respectively, proving that GPCN-2 could achieve photoelectric conversion more efficiently, which showed the more outstanding photocatalytic performance. The quantity of released hydrogen remained at 13.15 mmol g$^{-1}$ after five cycles as shown in Figure 4d, which indicated that GPCN-2 catalyst possessed favor-

able stability. Its splendid stability enables it to be reused, which greatly improves the utilization rate of the catalysts for practical applications.

$$AQE = \frac{2 \times \text{amount of evolved } H_2 \text{ molecules}}{\text{Number of incident photons}} \tag{3}$$

The dye elimination experiments were particularly analyzed as well for further assessment of photocatalytic activity. Rhodamine B solution with an original concentration of 20 mg $L^{-1}$ was selected as the simulated target contaminant. The absorption spectrogram of RhB at wavelengths between 350 and 750 nm for different degradation times with the existence of GPCN-2 is displayed in Figure 5a. It can be observed that the maximum absorption wavelength of RhB is approximately 554 nm, and the content of RhB in the reaction solution gradually decreased as the illumination time increased. Figure 5b suggests the degradation rates of RhB with different as-prepared catalysts under the same conditions. The results show that BCP presented the minimum adsorption rate (0.38%) and degradation rate (15.47%), which was much lower than those of PCN (16.63% of adsorption and 81.34% of degradation), resulting from the 3D network structure of PCN. Moreover, the adsorption rate of GPCN-2 increased to 37.16%, and the elimination rate of GPCN-2 (96.27%) was 1.18 times that of PCN, which manifested that the g-$C_3N_4$/rGO heterojunction and 3D porous structure synergistically enhanced the adsorption and photocatalytic elimination performance of the catalysts. In addition, the comparison of the present material with other reported materials in terms of light source, irradiation time and photocatalytic efficiency of RhB degradation are listed in Table S5. The results indicate that the prepared catalyst in this work was effective in the degradation of RhB. Likewise, Figure 5c presents the pseudo-first-order rate constant for RhB degradation (k) of different catalysts, which reflects the kinetics of the photocatalytic reaction for RhB degradation, fitted by the Langmuir–Hinshelwood (L-H) equation function model [63]. The value of k was obtained via the first order kinetic equation, as seen in Equation (4) [64]. $C_0$ and $C_t$ represent the residual concentration of RhB at times of zero and t, respectively. GPCN-2 displayed the biggest rate constant of 0.273 $min^{-1}$, which was 19.5 times and 1.95 times that of bulk g-$C_3N_4$ and g-$C_3N_4$, respectively, proving that the incorporation of rGO and g-$C_3N_4$ facilitated the removal of RhB by the catalysts. Nevertheless, the values of GPCN-1 and GPCN-3 were slightly lower than that of GPCN-2, which demonstrated that the presence of a large number of nanosheets might affect the reaction of the catalysts with the contaminants. Additionally, we performed five cycles of the RhB degradation with GPCN-2 in Figure S4, and the degradation rate was 91.79% in the fifth cycle, which was consistent with the hydrogen production test, indicating the stability of GPCN-2 catalyst.

$$\ln\left(\frac{C_t}{C_0}\right) = -kt \tag{4}$$

It is well known that free radicals, the unpaired electrons or groups formed by the breakage of covalent bonds in compounds, play important roles in catalytic processes [65]. To further explore the role of active particles during photocatalytic degradation, an active particle quenching assay and electron spin resonance (ESR) determination were performed. Figure 5d manifested the rate of degradation of RhB by GPCN-2 in the presence of different quenchers, including IPA, EDTA-$Na_2$ and BQ, which were the trapping agents of $^\bullet$OH, $h^+$ and $O_2^{\bullet-}$, respectively. The elimination rate of RhB was only 69.06% and 80.7% in the presence of BQ and EDTA-$Na_2$, which was 27.21% and 15.57% less than that with no quencher addition (96.27%), respectively, revealing that $O_2^{\bullet-}$ and $h^+$ played crucial role in the oxidative photocatalytic degradation. However, the degradation rate of RhB presented a slight decrease under IPA, indicating a minor involvement of $^\bullet$OH. Furthermore, the mechanisms of the active particle formation are represented in the following formulas:

$$\text{GPCN catalyst} + h\nu \rightarrow h^+ + e^- \tag{5}$$

$$e^- + O_2 \rightarrow O_2^{\bullet-} \tag{6}$$

$$\text{rhodamine B} + h^+ \rightarrow \text{products} \tag{7}$$

$$\text{rhodamine B} + O_2^- \rightarrow \text{products} \tag{8}$$

$$2e^- + O_2 + 2H^+ \rightarrow H_2O_2 \tag{9}$$

$$H_2O_2 + e^- \rightarrow {}^\bullet OH + OH^- \tag{10}$$

$$H_2O_2 + O_2^{\bullet-} \rightarrow {}^\bullet OH + OH^- + O_2 \tag{11}$$

$$h^+ + OH^- \rightarrow {}^\bullet OH \tag{12}$$

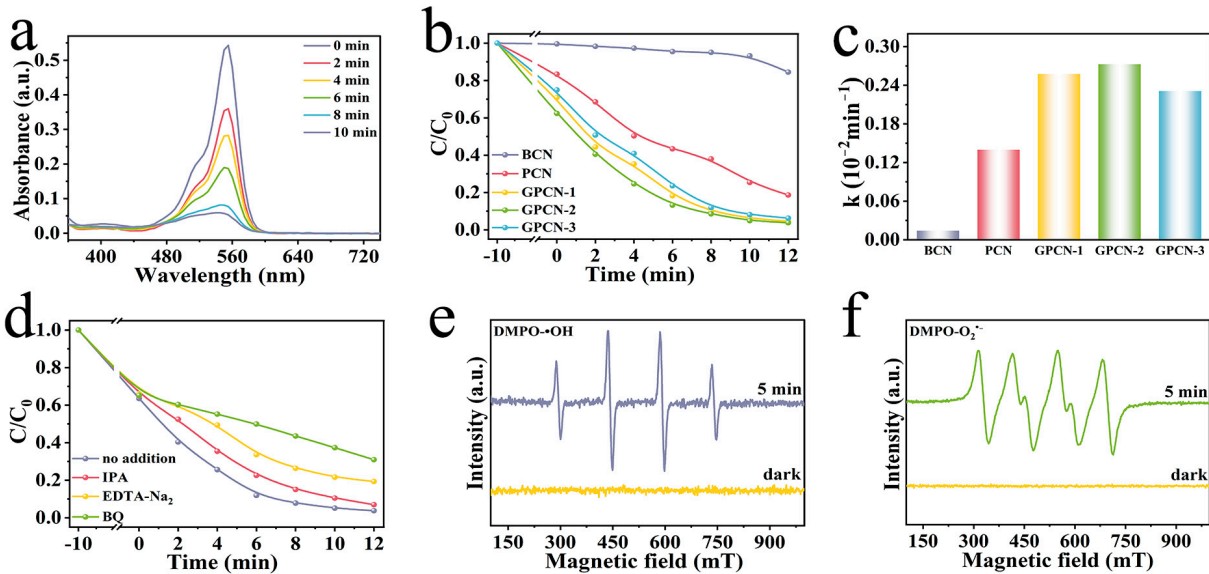

**Figure 5.** (**a**) Change in UV-vis spectra of RhB in the existence of GPCN-2; (**b**) elimination of RhB and (**c**) the degradation rate constants by the as-prepared catalysts; (**d**) degradation of RhB by GPCN-2 with existence of active species scavengers; and ESR spectrum of (**e**) DMPO-$O_2^{\bullet-}$ and (**f**) DMPO-$^\bullet$OH.

The consequences of ESR measurements are suggested in Figure 5e,f, which mutually corroborates the results of the free radical quenching assay. Both $^\bullet$OH and $O_2^{\bullet-}$ presented a strong response signal under irradiation but no response in the dark, certifying the significant effectiveness of $^\bullet$OH and $O_2^{\bullet-}$.

### 2.4. Mechanism Analysis of Photocatalytic Performance

On the basis of the above characterization analysis and photocatalysis experiments, a possible enhanced mechanism of the catalysts in photocatalytic performance was proposed as follows. Firstly, the typical 3D network structure and graphene nanosheets enlarge the superficial area of the catalysts, which results in more active sites and is conducive to adsorption and photocatalytic degradation. In the next place, the reduction of graphene oxide provides an abundance of active functional groups, which combine with organic matter to facilitate photocatalytic reactions. Finally, the g-C$_3$N$_4$/rGO heterojunction effectively not only promotes the separation and migration of photogenerated charges but also aggrandizes the lifetime of free electrons, providing more possibilities for electrons to participate in the RhB degradation process. The details of the specific chemical reactions in the photocatalytic decomposition of water and degradation of RhB are depicted in Figure 6. The charges on the valence bands (VB) of PCN transition to conduction bands (CB) under visible light exposure, transferring to rGO through the heterojunction. The photogenerated charges of rGO not only directly participate in hydrogen evolution, but also promote the formation of reactive free radicals and indirectly participate in the elimination of RhB. In addition, the resulting holes can directly take part in the removal of RhB. Hence, the

establishment of the heterojunctions effectively restrains the recombination of electron-hole pairs, providing more dissociative charges. Moreover, the 3D porous structure endows more adsorption sites on the catalysts, synergizing with the heterojunctions to enhance photocatalytic performance.

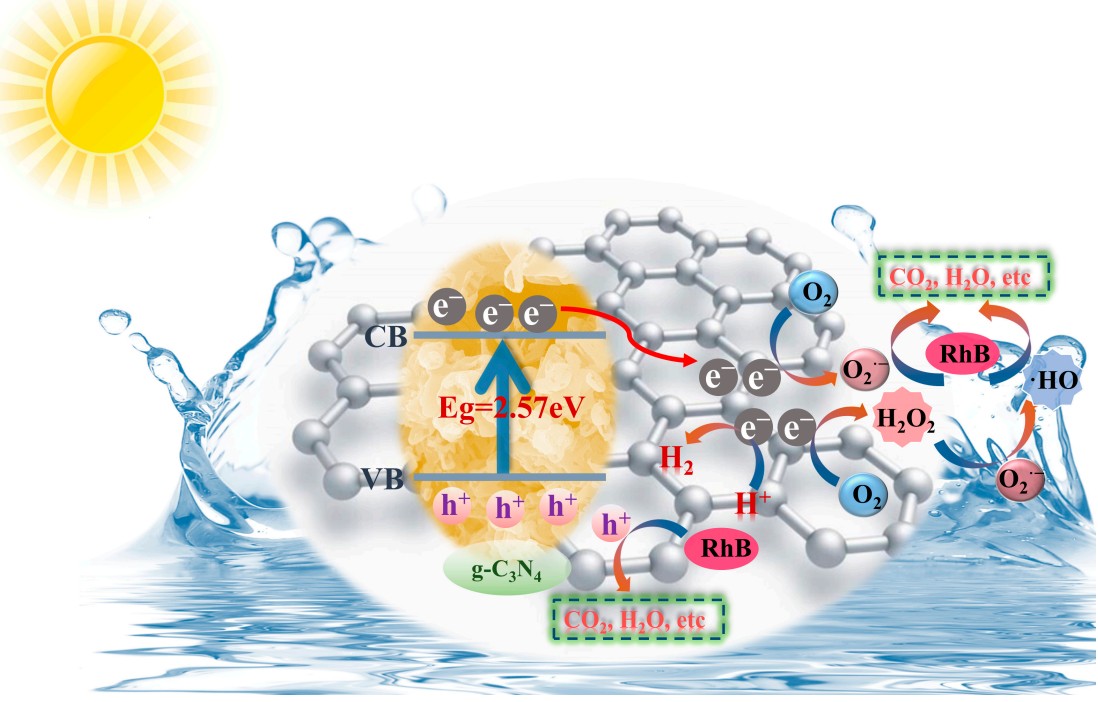

**Figure 6.** Possible enhanced mechanism of $H_2$ evolution and RhB elimination by GPCN-2.

## 3. Materials and Methods

### 3.1. Materials and Reagents

Melamine (Chemically pure, 99%), cyanuric acid (Chemically pure,98%), rhodamine B (RhB) (Analytical reagent), ethylene diamine tetraacetic acid disodium salt (EDTA-Na$_2$) (Guaranteed reagent, 99%), isopropanol (IPA) (Chromatographically pure, ≥99.9%) and benzoquinone (BQ) (Chemically pure, 99%) were obtained from Macklin Biochemical Co., Ltd. (Shanghai, China). Graphene oxide (GO) (Chemically pure, 99%) was acquired from XFNANO Materials Tech Co., Ltd. (Nanjing, China). All pharmaceutical products were analytically pure and had no purification.

### 3.2. Preparation of Catalysts

The preparation processes of the catalysts are shown in Scheme 1. The 3D g-C$_3$N$_4$ was prepared via a facile one-step calcination method. Firstly, 2.58 g melamine and 2.56 g cyanuric acid were dispersed in 100 mL H$_2$O and stirred overnight to achieve adequate mixing, and melamine–cyanuric acid supramolecular (MCS) was acquired. Subsequently, the above mixture was decanted into the plate and dried at 60 °C for 8 h. The obtained power was ground uniformly with a mortar and pestle, and the polymeric g-C$_3$N$_4$ (denoted as PCN) was obtained via annealing at 550 °C for 4 h in the muffle furnace. In addition, bulk g-C$_3$N$_4$ (named as BCN) was prepared using the same procedure without the addition of cyanuric acid.

In addition, the g-C$_3$N$_4$/rGO heterojunctions were obtained by reductive calcination in a tube furnace. A certain quantity of GO solution was added to a mixed solution of cyanuric acid and melamine during the synthetic process, and the resultant mixture was stirred, dried and annealed as PCN. Finally, after annealing at 350 °C for 2 h, the powder was calcined in a nitrogen atmosphere to reduce GO to rGO [66]. The g-C$_3$N$_4$/rGO

composite catalysts with 0.1%, 0.3% and 0.5% of GO were labelled as GPCN-1, GPCN-2 and GPCN-3, respectively.

The concrete characterizations and instrumental methods for the as-prepared catalysts are listed in the Supporting Information.

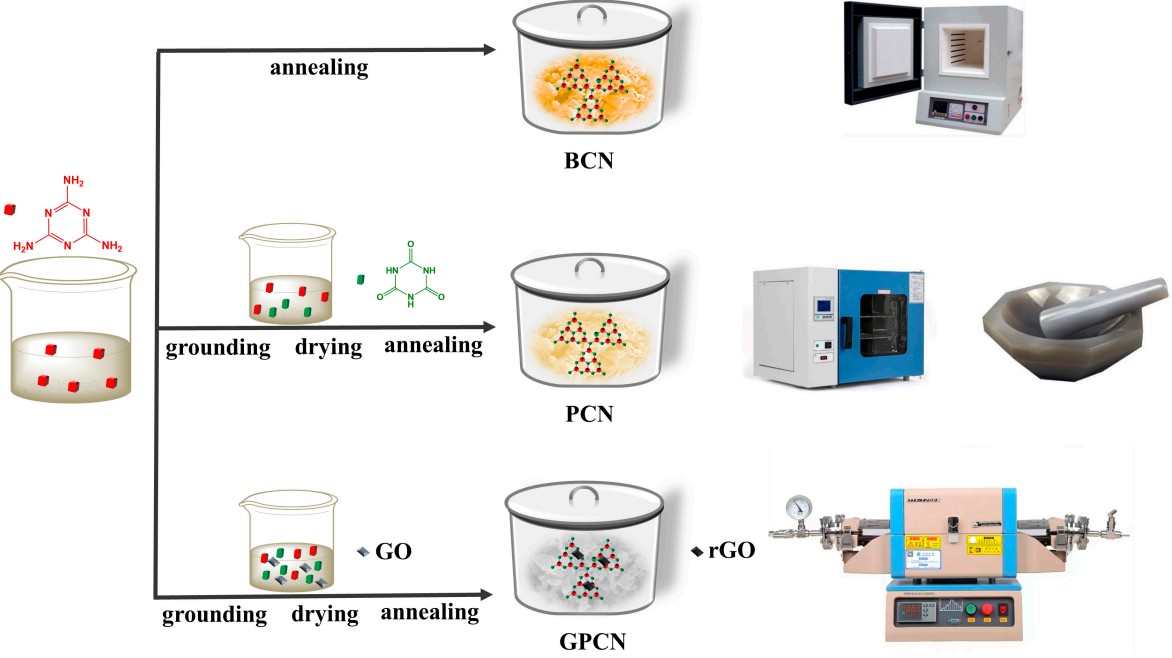

**Scheme 1.** Schematic diagram of the preparation process of the BCN, PCN and GPCN.

*3.3. Photocatalytic Performance Valuation*

The photocatalytic performance of the catalysts was evaluated from the perspective of hydrogen evolution and RhB degradation in this work. Photocatalytic degradation tests were carried out in a photocatalytic reactor (YZ-GHX-A, Yanzheng Experimental Instrument Co., Ltd., Shanghai, China). Photocatalytic hydrogen production tests were conducted using a photocatalytic analysis system (Labsolar 6A, Perfectlight Technology Co., Ltd., Beijing, China) with a 300 W Xenon lamp under visible light radiation ($\lambda \geq 420$ nm). Specific experimental details related to photocatalysis are mentioned in the supporting materials.

**4. Conclusions**

In summary, we simply synthesized 3D porous g-$C_3N_4$/rGO composite catalysts using cyanuric acid, melamine and GO as the raw materials. A series of characterization results showed that the heterostructure construction had no effect on the morphological characteristics of the catalysts. Meanwhile, the GPCN catalyst reduced the recombination of electron-hole pairs and improved the migration efficiency of photogenerated charge carriers. Moreover, the GPCN-2 catalyst presented the distinguished photocatalytic hydrogen evolution of 13.11 mmol $g^{-1}$ in 3 h, and GPCN-2 achieved 96.27% elimination of RhB in 12 min, which showed the most excellent performance among the as-prepared samples. The 3D structure and graphene nanosheets increase surface adsorption and the g-$C_3N_4$/rGO heterojunction promotes the separation of photogenerated charge carriers, which synergistically enhances the photocatalytic ability of the catalysts. Furthermore, a potential mechanism for enhancement was proposed. It can be found that the focus of future research will be on the synthesis of more environmentally friendly and efficient g-$C_3N_4$-based catalysts through the related research in this work, further exploring its potential value to expand its application in other catalytic fields. Overall, this work affords a new strategy for the facile fabrication of highly efficient catalysts, which is conducive to the development of clean energy and the alleviation of water pollution.

**Supplementary Materials:** The following supporting information can be downloaded at: https://www.mdpi.com/article/10.3390/catal13071079/s1, Table S1: The comparison of photocatalytic performance with reported 2D g-$C_3N_4$-based materials and 3D rGO/g-$C_3N_4$, Table S2: Specific surface areas and total pore volumes of BCN, PCN, GPCN-1, GPCN-2 and GPCN-3, Table S3: The contents of C and N in PCN and GPCN-2 catalysts according to the XPS measurement, Table S4: Lifetime profile and corresponding carrier dynamics information of PCN and GPCN-2, Table S5: The comparison of photocatalytic RhB degradation performance with reported g-$C_3N_4$-based photocatalysts, Figure S1: Schematic illustration of the preparation routes for the GPCN, Figure S2: The plots of transformed Kubelka–Munk function versus light energy of GPCN-2, Figure S3: Transient photocurrent of PCN and GPCN-2, Figure S4: Photocatalytic stability of GPCN-2. References [67–72] are cited in the Supplementary Materials.

**Author Contributions:** Conceptualization, H.S., T.N. and D.L.; methodology, C.Z.; software, C.L.; validation, H.S., T.N. and D.L.; formal analysis, M.C.; investigation, M.W., J.W. and S.J.; resources, D.L.; data curation, T.N.; writing—original draft preparation, D.L.; writing—review and editing, H.S., M.W. and J.W.; visualization, T.N.; supervision, D.L.; project administration, H.S.; funding acquisition, H.S., T.N. and D.L. All authors have read and agreed to the published version of the manuscript.

**Funding:** This research was funded by the National Natural Science Foundation of China (grant numbers 12105357), Starting Research Fund of Xinxiang Medical University (grant numbers XYB-SKYZZ201911), and Science and Technology Development Project of Henan Province (grant numbers 222102320316).

**Data Availability Statement:** The data presented in this study are available on request from the corresponding author.

**Conflicts of Interest:** The authors declare no conflict of interest.

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
