# Peer review of "Facile Synthesis of 3D Interconnected Porous g-C3N4/rGO Composite for Hydrogen Production and Dye Elimination"

_catalysts, doi:10.3390/catal13071079_

Round 1
Reviewer 1 Report
The manuscript titled “Facile synthesis of 3D interconnected porous g-C3N4/rGO composite for hydrogen production and dye elimination” is well written and explains the results well. Before publication for Catalyst, the manuscript needs some important changes which are given as.
· Abstract is too general, add some experimental data.
· Introduction, add industrial application of RhB and then their side effects.
· Introduction, explore more about the benefits of photocatalysis over other methods. Authors can review some literature e.g. https://doi.org/10.3390/molecules28031140, https://doi.org/10.3390/polym15030553,
· Add more information about the role of rGO in photocatalytic reaction.
· What is the novelty of your work as the material is already reported in many research articles. In the introduction, please explain the novelty of your work.
· The author may add the purity of the chemicals in materials and method section.
· In materials and methods section, Author may add citation for the method for synthesis of photocatalyst.
· Figure 2 (e, f) is wrong, how can C1s peak appear at around 400 eV? The author may check it again and explain.
· What is the bandgap value of GPCN?
· How GPCN-2 produce higher H2 as compared to GPCN-3?
· What is the sacrificial reagent used for hydrogen production. Authors, please mention this in text.
· Add a comparison of present material with other reported material with respect RhB dye such as irradiation time, photocatalytic efficiency, reaction rate, source of light etc.
· Add futures perspective for the present material in the conclusion section.
· Add more morphological, spectroscopic results in the conclusion section.

Author Response
To Mr. Laszlo-Szabolcs Kohr
Assistant Editor,
Catalysts
Dear Mr. Laszlo-Szabolcs Kohr
Ms. Ref. No.: catalysts-2473893
Title: Facile synthesis of 3D interconnected porous g-C3N4/rGO composite for hydrogen production and dye elimination
Thank you and reviewers very much for your kind efforts and the constructive comments and suggestions upon which we have revised the manuscript (yellow highlighted) as summarized as follows:
Response to reviewers:
Reviewer #1:
The manuscript titled “Facile synthesis of 3D interconnected porous g-C3N4/rGO composite for hydrogen production and dye elimination” is well written and explains the results well. Before publication for Catalyst, the manuscript needs some important changes which are given as.
Comment 1-1: Abstract is too general, add some experimental data.
Response to Reviewer comment No. 1-1:
-Thanks and as suggested, the results of structural characterization and photochemical characterization were summarized in the abstract section, as shown in lines 20-23.
Comment 1-2: Introduction, add industrial application of RhB and then their side effects.
Response to Reviewer comment No. 1-2:
-Thanks and as suggested, the industrial application of RhB and its side effects were added in the manuscript, as shown in lines 42-45.
Comment 1-3: Introduction, explore more about the benefits of photocatalysis over other methods. Authors can review some literature e.g.
https://doi.org/10.3390/molecules28031140,
https://doi.org/10.3390/polym15030553,
Response to Reviewer comment No. 1-3:
-Thanks and as suggested, the benefits of photocatalysis have been added, and we also cited these references in the manuscript, as shown in lines 49-51.
Comment 1-4: Add more information about the role of rGO in photocatalytic reaction.
Response to Reviewer comment No. 1-4:
-Thanks and as suggested, more information about the role of rGO in photocatalytic reaction was summarized in the induction of the manuscript, as shown in lines 95-97.
Comment 1-5: What is the novelty of your work as the material is already reported in many research articles. In the introduction, please explain the novelty of your work.
Response to Reviewer comment No. 1-5:
-Thanks and as suggested, we prepared g-C3N4 with a 3D interconnected porous structure, which provided larger surface areas and more active sites. The prepared GPCN catalysts exhibit excellent adsorption capacity of RhB, efficient utilization of incident illumination and rapid charge transfer due to the 3D structure. The novelty of this work was shown in lines 77-81.
Comment 1-6: The author may add the purity of the chemicals in materials and method section.
Response to Reviewer comment No. 1-6:
-Thanks and as suggested, the purity of the chemicals was added in the materials and method section, as shown in lines 110-114.
Comment 1-7: In materials and methods section, Author may add citation for the method for synthesis of photocatalyst.
Response to Reviewer comment No. 1-7:
-Thanks and as suggested, the related reference was cited in the manuscript, as shown in line 131.
Comment 1-8: Figure 2 (e, f) is wrong, how can C1s peak appear at around 400 eV? The author may check it again and explain.
Response to Reviewer comment No. 1-8:
-Thanks and sorry for the confusion, we have corrected it in the manuscript, as shown in Figure 2e and Figure 2f.
Comment 1-9: What is the bandgap value of GPCN?
Response to Reviewer comment No. 1-9:
-Thanks and as suggested, the bandgap value of GPCN-2 was 2.53 eV, which was calculated in Figure S2.
Figure S2. The plots of transformed Kubellka-Munk function versus light energy of GPCN-2.
Comment 1-10: How GPCN-2 produce higher H2 as compared to GPCN-3?
Response to Reviewer comment No. 1-10:
-Thanks and as suggested, the hydrogen production of GPCN-3 (10.87 mmol g-1) decreased with the increase of rGO, which may be due to the reason that the excess rGO hindered the capture and absorption of light by g-C3N4. The related explanation was shown in lines 298-300.
Comment 1-11: What is the sacrificial reagent used for hydrogen production. Authors, please mention this in text.
Response to Reviewer comment No. 1-11:
-Thanks and as suggested, the sacrificial agent of hydrogen production is triethanolamine, as mentioned in line 283.
Comment 1-12: Add a comparison of present material with other reported material with respect RhB dye such as irradiation time, photocatalytic efficiency, reaction rate, source of light etc.
Response to Reviewer comment No. 1-12:
-Thanks and as suggested, the comparison of the present material with other reported materials in terms of light source, irradiation time and photocatalytic efficiency of RhB degradation were listed in Table S5.
Comment 1-13: Add futures perspective for the present material in the conclusion section.
Response to Reviewer comment No. 1-13:
-Thanks and as suggested, the futures perspective for the present material was added in the conclusion section, as shown in lines 400-403.
Comment 1-14: Add more morphological, spectroscopic results in the conclusion section.
Response to Reviewer comment No. 1-14:
-Thanks and as suggested, the results of morphology and spectroscopy were described in the conclusion section, as shown in lines 390-394.
Reviewer #2:
In this article, C. Zhao et al. reported the facile preparation of 3D structure-based GCN/rGO heterostructure for hydrogen production and dye elimination applications. The authors have a solid background in photoelectrochemical-based applications. Although the GCN/rGO heterostructure-based hydrogen and dye degradation reports have been published, however, the concept theme of making 3D porous GCN and its hybridization with rGO is quite interesting. The results were really nice and acceptable, and a suitable mechanism was described. However, there are some issues that authors need to be rectified and answer well before it can be published.
Comment 2-1: Please expand the 3D in the abstract, and whenever the authors want to use the notations, please explain what it was for better understanding, like, for instance, three-dimensional (3D). Please check the entire manuscript.
Response to Reviewer comment No. 2-1:
-Thanks and as suggested, three-dimensional (3D) was modified in the abstract, as shown in line 18. All abbreviations have been explained in the manuscript.
Comment 2-2: It Stands out because of its stable structure; please modify the sentence and write something after the word stands out for what? For a complete sentence. Some part of the manuscript needs some grammatical and English correction. Please proofread the entire manuscript.
Response to Reviewer comment No. 2-2:
-Thanks and sorry for the confusion. The sentence “stands out because of its stable structure” was modified to “is widely used by researchers due to the stable structure” as shown in line 60. The full text was also checked for grammar, and the changes were highlighted in yellow.
Comment 2-3: The title itself resembles the dual application of your active catalyst (GPCN) for hydrogen production and dye elimination. However, authors have mainly focused on the dye-related advantages/limitations only in the introduction. Please also include and give equal priority to hydrogen production.
Response to Reviewer comment No. 2-3:
-Thanks and as suggested, the advantages of hydrogen production were added in the introduction, as shown in lines 37-41.
Comment 2-4: What is the structural novelty of your active catalyst?
Response to Reviewer comment No. 2-4:
-Thanks and as suggested, we prepared g-C3N4 with a 3D hollow porous structure, which provided larger surface areas and more active sites for the catalysts. The prepared GPCN catalysts exhibited excellent adsorption capacity of RhB dye, efficient utilization of illumination and rapid charge transfer due to the 3D interconnected structure. The novelty of this work was shown in lines 77-81.
Comment 2-5: What is PCN? Please expand it before using it everywhere in line 107, page 3.
Response to Reviewer comment No. 2-5:
-Thanks and as suggested, PCN is the abbreviation of polymeric g-C3N4, as explained in lines 123-124.
Comment 2-6: At what temperature was GO reduced to rGO? Include in the manuscript.
Response to Reviewer comment No. 2-6:
-Thanks and as suggested, GO was reduced to rGO at 350 °C for 2 h, which was displayed in line 130.
Comment 2-7: For hydrogen generation via PEC experiments, most of the researchers used 1 sun intensity of light with 100 mW cm-2 rather than 30 mW cm-2. Please explain.
Response to Reviewer comment No. 2-7:
-Thanks and sorry for the confusion. A 500 W metal halide lamp with an average light intensity of 30 mW cm-2 was used for RhB degradation. While a 300 W Xe lamp with an average light intensity of 100 mW cm-2 was used for hydrogen generation. The details were displayed in the supporting materials.
Comment 2-8: Please provide the linear sweep voltammograms and photocurrent density spectrum cycles of the corresponding photocatalyst in the supplementary information.
Response to Reviewer comment No. 2-8:
-Thanks and as suggested, we supplemented the transient photocurrent spectra of PCN and GPCN-2 in Fig. S3, which were consistent with EIS results. The maximum intensity of GPCN-2 (16.8 μA) was 3.05 times of PCN (5.5 μA). Additionally, PL and TRPL were also conducted to evaluate the reorganization of photogenerated charge carriers in the catalysts.
Figure S3. Transient photocurrent of PCN and GPCN-2.
Comment 2-9: Why did the author choose 0.1 M Na2SO4 only? Because at higher molar ratios < 0.5 M, there will be more probability of active charge interaction and interactions and fast flow of electrons. Please explain.
Response to Reviewer comment No. 2-9:
-Thanks and sorry for the confusion. For EIS experiments, we chose 0.1 M Na2SO4 and ensured that all samples were characterized under the same conditions to compare their electron transport efficiency. In addition, the similar experiment condition of EIS can be found in previous studies [Appl. Catal. B: Environ., 2016, 183, 133-141; Appl. Catal. B: Environ., 2019, 242, March, 92-99; Chem. Eng. J., 2023, 455, 140570; Chem. Eng. J., 2023, 455, 140570].
Comment 2-10: In my opinion, compared to ITO, FTO is more reliable and widely used in PEC-based H2 production and dye degradation. Please explain. How did the material was coated on the ITO substrates, and how did the authors achieve the desired thickness? How about the active catalyst adhesiveness? Any binder that they used? Explained detailedly in the supplementary information.
Response to Reviewer comment No. 2-10:
-Thanks and sorry for the confusion. We conducted hydrogen production and RhB degradation by photocatalysis without using ITO or FTO. The ITO was used for the EIS measurement, and the details were displayed in supporting materials.
Comment 2-11: Please compare your research material with the other 2D, polymer-based materials and enrich your research novelty, as some of the references are not up to date. Here are some of the references I suggest to authors that could be helpful. https://doi.org/10.1016/j.jece.2019.103289, https://doi.org/10.1007/s12274-023-5472-x, https://doi.org/10.1021/acsami.3c00239, 10.1016/j.apcatb.2022.121941, https://doi.org/10.1016/j.jallcom.2022.164500
Response to Reviewer comment No. 2-11:
-Thanks and as suggested, we added the comparison of photocatalytic performance with reported 2D g-C3N4-based materials and 3D rGO/g-C3N4 in this work, as listed in Table S1.
Again, we sincerely appreciate the editor and reviewers’ comments. We hope that the revision has fully addressed the issues brought up in the review.
Sincerely yours,
Dong Liu

Reviewer 2 Report
In this article, C. Zhao et al. reported the facile preparation of 3D structure-based GCN/rGO heterostructure for hydrogen production and dye elimination applications. The authors have a solid background in photoelectrochemical-based applications. Although the GCN/rGO heterostructure-based hydrogen and dye degradation reports have been published, however, the concept theme of making 3D porous GCN and its hybridization with rGO is quite interesting. The results were really nice and acceptable, and a suitable mechanism was described. However, there are some issues that authors need to be rectified and answer well before it can be published.
1) Please expand the 3D in the abstract, and whenever the authors want to use the notations, please explain what it was for better understanding, like, for instance, three-dimensional (3D). Please check the entire manuscript.
2) It Stands out because of its stable structure; please modify the sentence and write something after the word stands out for what? For a complete sentence. Some part of the manuscript needs some grammatical and English correction. Please proofread the entire manuscript.
3) The title itself resembles the dual application of your active catalyst (GPCN) for hydrogen production and dye elimination. However, authors have mainly focused on the dye-related advantages/limitations only in the introduction. Please also include and give equal priority to hydrogen production.
4) What is the structural novelty of your active catalyst?
5) What is PCN? Please expand it before using it everywhere in line 107, page 3.
6) At what temperature was GO reduced to rGO? Include in the manuscript.
7) For hydrogen generation via PEC experiments, most of the researchers used 1 sun intensity of light with 100 mW cm-2 rather than 30 mW cm-2. Please explain.
8) Please provide the linear sweep voltammograms and photocurrent density spectrum cycles of the corresponding photocatalyst in the supplementary information.
9) Why did the author choose 0.1 M Na2SO4 only? Because at higher molar ratios < 0.5 M, there will be more probability of active charge interaction and interactions and fast flow of electrons. Please explain.
10) In my opinion, compared to ITO, FTO is more reliable and widely used in PEC-based H2 production and dye degradation. Please explain. How did the material was coated on the ITO substrates, and how did the authors achieve the desired thickness? How about the active catalyst adhesiveness? Any binder that they used? Explained detailedly in the supplementary information.
11) Please compare your research material with the other 2D, polymer-based materials and enrich your research novelty, as some of the references are not up to date. Here are some of the references I suggest to authors that could be helpful. https://doi.org/10.1016/j.jece.2019.103289, https://doi.org/10.1007/s12274-023-5472-x, https://doi.org/10.1021/acsami.3c00239, 10.1016/j.apcatb.2022.121941, https://doi.org/10.1016/j.jallcom.2022.164500
Author Response

(The authors gave the same response as above.)

Round 2
Reviewer 1 Report
The author has improved the quality of the manuscript. I recommend this for publication in its current form.